# Effective and Efficient Vote Attack on Capsule Networks

**Jindong Gu**[1,4]**, Baoyuan Wu**[2,3]**, Volker Tresp**[1,4]
University of Munich, Germany[1]
The Chinese University of Hong Kong, Shenzhen, China[2]
Shenzhen Research Institute of Big Data, Shenzhen, China[3]
Corporate Technology, Siemens AG, Munich, Germany[4]
`jindong.gu@outlook.com, wubaoyuan@cuhk.edu.cn, volker.tresp@siemens.com`

## Abstract

Standard Convolutional Neural Networks (CNNs) can be easily fooled by images with small quasi-imperceptible artificial perturbations. As alternatives to CNNs, the recently proposed Capsule Networks (CapsNets) are shown to be more robust to white-box attacks than CNNs under popular attack protocols. Besides, the class-conditional reconstruction part of CapsNets is also used to detect adversarial examples. In this work, we investigate the adversarial robustness of CapsNets, especially how the inner workings of CapsNets change when the output capsules are attacked. The first observation is that adversarial examples misled CapsNets by manipulating the votes from primary capsules. Another observation is the high computational cost, when we directly apply multi-step attack methods designed for CNNs to attack CapsNets, due to the computationally expensive routing mechanism. Motivated by these two observations, we propose a novel vote attack where we attack votes of CapsNets directly. Our vote attack is not only effective but also efficient by circumventing the routing process. Furthermore, we integrate our vote attack into the detection-aware attack paradigm, which can successfully bypass the class-conditional reconstruction based detection method. Extensive experiments demonstrate the superior attack performance of our vote attack on CapsNets.

## 1 Introduction

A hardly perceptible small artificial perturbation can cause Convolutional Neural Networks (CNNs) to misclassify an image. Such vulnerability of CNNs can pose potential threats to security-sensitive applications, *e.g.*, face verification (Sharif et al., 2016) and autonomous driving (Eykholt et al., 2018). Besides, the existence of adversarial images demonstrates that the object recognition process in CNNs is dramatically different from that in human brains. Hence, the adversarial examples have received increasing attention since it was introduced (Szegedy et al., 2014; Goodfellow et al., 2015).

Many works show that network architectures play an important role in adversarial robustness (Madry et al., 2018; Su et al., 2018; Xie & Yuille, 2020; Guo et al., 2020). As alternatives to CNNs, Capsule Networks (CapsNets) have also been explored to resist adversarial images since they are more biologically inspired (Sabour et al., 2017). The CapsNet architectures are significantly different from those of CNNs. Under popular attack protocols, CapsNets are shown to be more robust to white-box attacks than counter-part CNNs (Hinton et al., 2018; Hahn et al., 2019). Furthermore, the reconstruction part of CapsNets is also applied to detect adversarial images (Qin et al., 2020).

In image classifications, CapsNets first extract primary capsules from the pixel intensities and transform them to make votes. The votes reach an agreement via an iterative routing process. It is not clear how these components change when CapsNets are attacked. By attacking output capsules directly, the robust accuracy of CapsNets is $17.3\%$, while it is reduced to $0$ on the counter-part CNNs in the same setting. Additionally, it is computationally expensive to apply multi-step attacks (*e.g.*, PGD (Madry et al., 2018)) to CapsNets directly, due to the costly routing mechanism. The two observations motivate us to propose an effective and efficient vote attack on CapsNets.

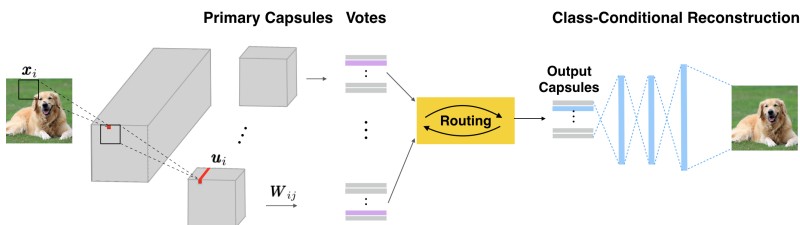

Figure 1: The overview of Capsule Networks: the CapsNet architecture consists of four components, *i.e.*, primary capsule extraction, voting, routing, and class-conditional reconstruction.

The contributions of our work can be summarised as follows: 1). We investigate the inner working changes of CapsNets when output capsules are attacked; 2). Motivated by the findings, we propose an effective and efficient vote attack; 3). We integrate the vote attack in the detection-aware attack to bypass class-conditional reconstruction based adversarial detection. The next section introduces background knowledge and related work. Sec. 3 and 4 investigate capsule attack and introduce our vote attack, respectively. The last two sections show experiments and our conclusions.

## 2 BACKGROUND KNOWLEDGE AND RELATED WORK

**Capsule Networks** The overview of CapsNets is shown in Figure 1. CapsNets first extract primary capsules $\boldsymbol{u}_i$ from the input image $\boldsymbol{x}$ with pure convolutional layers (or CNN backbones). Each primary capsule $\boldsymbol{u}_i$ is then transformed to make votes for high-level capsules. The **voting process**, also called transformation process, is formulated as

$$\hat{\boldsymbol{u}}_{j|i} = \boldsymbol{W}_{ij}\boldsymbol{u}_i. \tag{1}$$

Next, a dynamic routing process is applied to identify weights $c_{ij}$ for the votes $\hat{\boldsymbol{u}}_{j|i}$, with $i \in \{1, 2, \dots, N\}$ corresponding to indices of primary capsules and $j \in \{1, 2, \dots, M\}$ to indices of high-level capsules. Specifically, the **routing process** iterates over the following three steps

$$\boldsymbol{s}_j^{(t)} = \sum_i^N c_{ij}^{(t)}\hat{\boldsymbol{u}}_{j|i}, \qquad \boldsymbol{v}_j^{(t)} = g(\boldsymbol{s}_j^{(t)}), \qquad c_{ij}^{(t+1)} = \frac{\exp(b_{ij} + \sum_{r=1}^t \boldsymbol{v}_j^{(r)}\hat{\boldsymbol{u}}_{j|i})}{\sum_k \exp(b_{ik} + \sum_{r=1}^t \boldsymbol{v}_k^{(r)}\hat{\boldsymbol{u}}_{k|i})}, \tag{2}$$

where the superscript $t$ indicates the index of iterations starting from 1, and $g(\cdot)$ is a squashing function (Sabour et al., 2017) that maps the length of the vector $\boldsymbol{s}_j$ into the range of $[0, 1)$. The $b_{ik}$ is the log prior probability. Note that the routing process is the most expensive part of CapsNets.

The final output capsules are computed as $\boldsymbol{v}_j = g(\sum_{i=1}^N c_{ij} * \hat{\boldsymbol{u}}_{j|i})$ where $c_{ij}$ is the output of the last routing iteration. The output capsules are represented by vectors, the length of which indicates the confidence of the entitys' existence. In the training phase, the class-conditional reconstruction net reconstructs the input image from the capsule corresponding to the ground-truth class $t$, *i.e.*, $\hat{\boldsymbol{x}} = r(\boldsymbol{v}_t)$. The reconstruction error $d(\boldsymbol{x}, \hat{\boldsymbol{x}}) = \|\hat{\boldsymbol{x}} - \boldsymbol{x}\|_2$ works as a regularization term. All above notations will be used across this manuscript.

To improve CapsNets (Sabour et al., 2017), various routing mechanisms have been proposed, such as (Hinton et al., 2018; Zhang et al., 2018; Hahn et al., 2019; Tsai et al., 2020). The advanced techniques of building CNNs or GNNs have also been integrated into CapsNets successfully. For example, the multi-head attention-based graph pooling is applied to replace the routing mechanism (Gu & Tresp, 2020b). The CNN backbones are applied to extract more accurate primary capsules (Rajasegaran et al., 2019; Phaye et al., 2018). To understand CapsNets, (Gu & Tresp, 2020a) investigates the contribution of dynamic routing to the input affine transformation robustness. This work focuses on its contribution to the adversarial robustness.

(Hinton et al., 2018; Hahn et al., 2019) demonstrated the high adversarial robustness of CapsNets. However, it has been shown in (Michels et al., 2019) that the robustness does not hold for all attacks. In addition, many defense strategies proposed for CNNs are circumvented by later defense-aware white-box attacks (Athalye et al., 2018). Given the previous research line, we argue that it is necessary to explore CapsNet architecture-aware attacks, before we give any claim on the robustness

of CapsNets. To the best of our knowledge, there is no attack specifically designed for CapsNets in current literature.

**Adversarial Attacks** Given the outputs $f(\boldsymbol{x})$ of an input in a CNN, attacks fool the model by creating perturbations to increase the loss $\mathcal{L}(f(\boldsymbol{x} + \boldsymbol{\delta}), \boldsymbol{y})$ where $\mathcal{L}(\cdot)$ is the standard cross-entropy loss and $\boldsymbol{\delta}$ indicates a $\ell_p$-bounded perturbation. The one-step *Fast Gradient Sign Method* (FGSM (Goodfellow et al., 2015)) creates perturbations as

$$\boldsymbol{\delta} = \epsilon \cdot \text{sign}(\nabla_{\boldsymbol{\delta}} \mathcal{L}(f(\boldsymbol{x} + \boldsymbol{\delta}), \boldsymbol{y})). \tag{3}$$

The multi-step *Projected Gradient Descent* (PGD (Madry et al., 2018)), is defined as

$$\boldsymbol{\delta} \leftarrow \text{clip}_\epsilon(\boldsymbol{\delta} + \alpha \cdot \text{sign}(\nabla_{\boldsymbol{\delta}} \mathcal{L}(f(\boldsymbol{x} + \boldsymbol{\delta}), \boldsymbol{y}))). \tag{4}$$

Other popular multi-step attacks also include *Basic Iteractive Method* (BIM (Kurakin et al., 2017)) *Momentum Iterative Method* (MIM (Dong et al., 2018)). Besides, C&W attack (Carlini & Wagner, 2017b) and Deepfool (Moosavi-Dezfooli et al., 2016) are popular strong attacks on the $\ell_2$-norm constraint.

**Adversarial Detection** Besides adversarial attack and defense (Madry et al., 2018; Chen et al., 2020; Li et al., 2020), adversarial detection has also received much attention (Xu et al., 2017; Ma et al., 2020). Many CNN-based adversarial detection methods were easily bypassed by constructing new loss functions (Carlini & Wagner, 2017a). Adversarial images are not easily detected. The most recent work (Qin et al., 2020) leverages the class-conditional reconstruction net of CapsNets to detect adversarial images.

Given any input $\boldsymbol{x}$, the predictions and the corresponding capsule are $f(\boldsymbol{x})$ and $\boldsymbol{V}$, respectively. The input is flagged as an adversarial image, if the reconstruction error is bigger than a pre-defined threshold $\|r(\boldsymbol{v}_p) - \boldsymbol{x}\|_2 > \theta$ where $p = \arg\max f(\boldsymbol{x})$ is the predicted class. The reconstruction net $r(\cdot)$ reconstructed the input from the capsule $\boldsymbol{v}_p$ of the predicted class. The choice of $\theta$ involves a trade-off between false positive and false negative detection rates. Instead of tuning this parameter, the work (Qin et al., 2020) simply sets it as the 95th percentile of benign validation distances. A strong detection-aware reconstructive attack is also proposed to verify the effectiveness of the proposed detection method in (Qin et al., 2020). The reconstructive attack is a two-stage optimization method where it first creates a perturbation $\boldsymbol{\delta}$ to fool the prediction as in Equation (5), and updates the perturbation further to reduce the reconstruction error as in Equation (6),

$$\boldsymbol{\delta} \leftarrow \text{clip}_\epsilon(\boldsymbol{\delta} + \alpha \cdot \beta \cdot \text{sign}(\nabla_{\boldsymbol{\delta}} \mathcal{L}(f(\boldsymbol{x} + \boldsymbol{\delta}), \boldsymbol{y}))), \tag{5}$$

$$\boldsymbol{\delta} \leftarrow \text{clip}_\epsilon(\boldsymbol{\delta} + \alpha \cdot (1 - \beta) \cdot \text{sign}(\nabla_{\boldsymbol{\delta}} \left\| r(\boldsymbol{v}_{f(\boldsymbol{x})}), \boldsymbol{x} \right\|_2), \tag{6}$$

where $\alpha$ is the step size, and $\beta$ is a hyper-parameter to balance the losses in the two stages.

## 3  CAPSULE ATTACK ON CAPSULE NETWORKS

**Attack Formulation.** In CNNs, under certain constraints, the adversary finds adversarial perturbation of an instance by maximizing the classification loss. In CapsNets, the length of output capsules corresponds to the output probability of the classes. Similarly, the adversarial perturbation can be obtained by first mapping the length of output capsules to logits $Z(\boldsymbol{x})_j = \log(\|\boldsymbol{v}_j\|_2)$ and solving the maximization problem in Equation (7). In this formulation, output capsules are attacked directly, which is called **Caps-Attack**.

$$\boldsymbol{\delta}^* = \underset{\boldsymbol{\delta} \in \mathcal{N}_\epsilon}{\arg\max} \, \mathcal{H}(Z(\boldsymbol{x} + \boldsymbol{\delta}), \boldsymbol{y}) = \mathcal{L}(\text{softmax}(Z(\boldsymbol{x} + \boldsymbol{\delta})), \boldsymbol{y}), \tag{7}$$

where $\mathcal{N}_\epsilon = \{\boldsymbol{\delta} : \|\boldsymbol{\delta}\|_p \leq \epsilon\}$ with $\epsilon > 0$ being the maximal perturbation. This optimization problem can be naturally solved using the algorithm designed for the attack against CNNs, such as FGSM (see Equation (3)) (Goodfellow et al., 2015) and PGD (see Equation (4)) (Madry et al., 2018).

**Analysis.** In CapsNets, the primary capsule $\boldsymbol{u}_i$ can make a positive or negative vote for the $j$-th class or abstain from voting. It depends on the relationship between $\boldsymbol{v}_j$ and $\hat{\boldsymbol{u}}_{j|i}$. The vote from $\boldsymbol{u}_i$ for the $j$-th class is positive if $\cos(\boldsymbol{v}_j, \hat{\boldsymbol{u}}_{j|i}) > 0$, otherwise negative if $\cos(\boldsymbol{v}_j, \hat{\boldsymbol{u}}_{j|i}) < 0$. The similarity value $\cos(\boldsymbol{v}_j, \hat{\boldsymbol{u}}_{j|i}) = 0$ corresponds to abstention of the primary capsules.

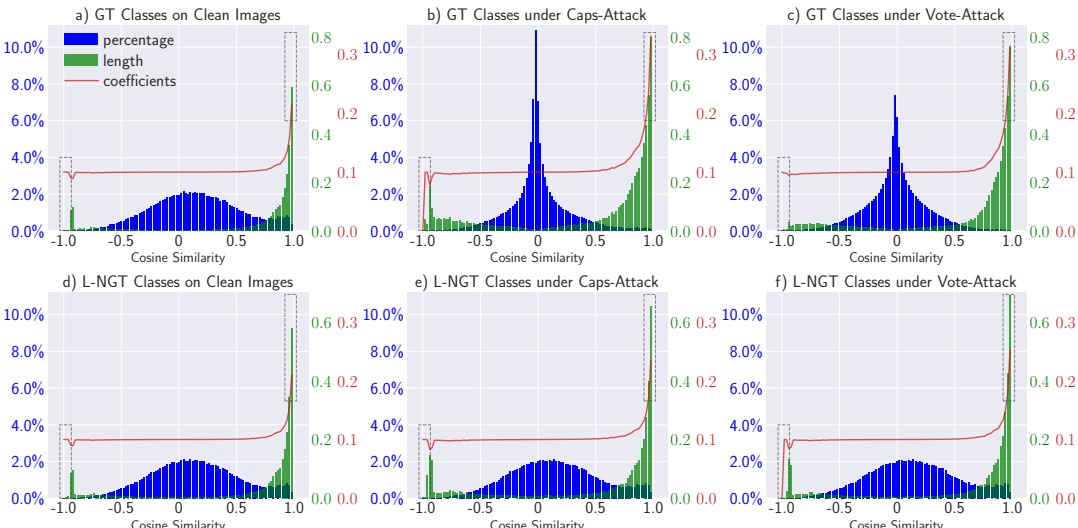

Figure 2: The left-to-right columns correspond to statistics of predictions on clean images, under Caps-Attack, and under Vote-Attack, respectively. The first row corresponds to the statistics on ground-truth classes, and the second row corresponds to the classes with the largest output probabilities that are not ground-truth (L-NGT) classes. In each subplot, the x-axis indicates the cosine similarity value between the vote $\hat{\boldsymbol{u}}_{j|i}$ and the output capsule $\boldsymbol{v}_j$. The blue histogram shows the percentage of votes falling in bins divided by the similarity values in x-axis. The green histogram corresponds to the strength of votes (the averaged length of the votes $\hat{\boldsymbol{u}}_{j|i}$). The red curve presents the averaged weight (*i.e.*, $c_{ij}$, see Equation (2)) of votes at each bin. Please refer to the main context for more in-depth analysis of this figure.

How do the votes change when CapsNets are attacked by adversarial images? We investigate this question with experiments and visualize the results. We firstly train a CapsNet with Dynamic Routing (DR-CapsNet) (Sabour et al., 2017) on the CIFAR10 dataset (Krizhevsky et al., 2009). With the standard well-trained DR-CapsNet (92.8% test accuracy), we classify all clean images in the test dataset and extract all votes $\hat{\boldsymbol{u}}_{j|i}$ and output capsules $\boldsymbol{v}_j$ of the ground-truth (GT) classes. We compute $\cos(\boldsymbol{v}_j, \hat{\boldsymbol{u}}_{j|i})$ in all classifications and split them into 100 equal-width bins in the range of $[-1, 1]$. In each bin, we compute the averaged length of all $\hat{\boldsymbol{u}}_{j|i}$ and average of all coupling coefficients $c_{ij}$ therein. Note that $c_{ij}$ identified by the routing process stands for the weights of the vote $\hat{\boldsymbol{u}}_{j|i}$. The results are visualized in Figure 2a. The majority of primary capsules make positive votes (more votes with positive similarity values in blue bins).

To obtain adversarial images, we apply PGD attack to the clean image classifications on the DR-CapsNet where 17.3% robust accuracy is obtained. Similarly, we extract corresponding information from the classifications of adversarial images on the ground-truth class and visualize the results in Figure 2b. The votes corresponding to $\cos(\boldsymbol{v}_j, \hat{\boldsymbol{u}}_{j|i}) \approx 0$ are invalid since they have only tiny impact on final prediction. The adversarial images make votes invalid by manipulating the votes and the weights of them. Concretely, the votes on adversarial images are $\hat{\boldsymbol{u}}'_{j|i}$. The voting weights identified by the routing process are $c'_{ij}$. Both are manipulated by adversarial images so that the output capsule $\boldsymbol{v}'_j = \sum_{i=1}^{N} c'_{ij} * \hat{\boldsymbol{u}}'_{j|i}$ is orthogonal to most votes $\hat{\boldsymbol{u}}'_{j|i}$. Namely, the adversarial images make the majority of votes invalid for the ground-truth class (the concentration of votes around the zero).

To understand how votes change on non-ground-truth classes, we also visualize the corresponding information on the classes with the **L**argest output probabilities that are **N**ot **G**round-**T**ruth classes (L-NGT classes) in Figure 2d and 2e. We mark differences between the two plots with dashed gray boxes. We can observe that the votes for L-NGT classes become stronger since both the coupling coefficients (the red line) and the strength of their positive votes (the green bins) become larger.

**Drawbacks.** The above analysis explains why the attack method originally designed for CNNs still works for CapsNets. The first drawback of Caps-Attack is its *limited effectiveness*. As will

be shown in later experiments, under the same attack method, CapsNets are much more robust than CNNs. Since the routing process is the main difference between CapsNets and CNNs, we attribute the higher robustness of CapsNets to the conjecture that the routing process obfuscates the gradients used to generate adversarial examples. One intuitive way to mitigate it is to approximate the routing process, *e.g.*, with Backward Pass Differentiable Approximation (BPDA) (Athalye et al., 2018). However, it is non-trivial to approximate the routing process with several routing iterations. The second drawback of Caps-Attack is the *low efficiency*. The widely used multi-step gradient-based attacks require many times forward and backward passes on the whole CapsNet to generate adversarial examples, *e.g.,* under PGD attack. Caps-Attack are computationally expensive due to the costly iterative routing mechanism of CapsNets.

## 4 VOTE ATTACK ON CAPSULE NETWORKS

The above two drawbacks of Caps-Attack inspire us that it is necessary to develop adversarial attack methods specifically for CapsNets, rather than directly applying the attack methods designed for CNNs to attack CapsNets. In this work, we propose to directly attack the votes (see Equation (8)) rather than the final output capsules of CapsNets, dubbed **Vote-Attack**. The behind rationale is that the vote $\hat{\boldsymbol{u}}_{j|i}$ exactly corresponds to the output class $j$, though it is an intermediate activation of CapsNets. Besides, when the votes from primary capsules are attacked, the corresponding weights (*i.e.*, $c_{ij}$, see Equation (2)) identified by the routing process will also be changed. Thus, the attacked votes could mislead the corresponding outputs of CapsNets.

Specifically, given an input-label pair $(\boldsymbol{x}, \boldsymbol{y})$, the N votes from primary capsules are $\hat{\boldsymbol{u}}_{-|i} = f_v^i(\boldsymbol{x})$ where $i \in \{1, 2, \ldots, N\}$. The average of the N votes is first computed and then squashed wtih the squashing fucntion $g(\cdot)$. The vector lengths of the squashed one correspond to output probabilities. Formally, the Vote-Attack on $\boldsymbol{x}$ is defined as

$$\boldsymbol{\delta}^* = \underset{\boldsymbol{\delta} \in \mathcal{N}_\epsilon}{\arg\max} \, \mathcal{H}(\log(g(\frac{1}{N}\sum_{i=1}^{N} f_v^i(\boldsymbol{x} + \boldsymbol{\delta}))), \boldsymbol{y}). \tag{8}$$

In the formulation above, we first average the votes and squash the averaged vote. There are two intuitive variants of the proposed Vote-attack. The one is to first squash their votes and then average the squashed votes. The other is to average the loss caused by all votes. Instead of opimizing on the loss computed on the squahed averaged vote, we can compute the loss of individual vote seperatedly and average them. More details about these two variants of our Vote-Attack can be found in Appendix A.

The maximization problem of Equation (8) can be approximately solved with popular attack method, *e.g.*, PGD attack. When PGD is taken as the underlying attack, the proposed Vote-Attack method can reduce the robust accuracy of DR-CapsNets from 17.3% (with Caps-Attack) to 4.83%.

Our Vote-Attack can also be extended to targeted attack by simply modifying the attack loss function of Equation (8) into $\boldsymbol{\delta}^* = \arg\max_{\boldsymbol{\delta} \in \mathcal{N}_\epsilon} l(\log(g(\frac{1}{N}\sum_{i=1}^{N} f_v^i(\boldsymbol{x} + \boldsymbol{\delta}))), \boldsymbol{t})$ where $\boldsymbol{t}$ is the target class.

**Analysis.** We also visualize the votes on the adversarial images created by our Vote-Attack. On the GT classes (see Figure 2c), our Vote-Attack increase the negative votes and decrease the positive votes, when compared to Caps-Attack in Figure 2b. On the L-NGT classes, the positive votes are strengthened further by our Vote-Attack, which leads to more misclassifications. See the difference in dashed gray boxes, where both the length of positive votes and the weights become larger (where the similarity values are about 1.0).

**Advantages.** It is interesting to find that the proposed Vote-Attack could alleviate the drawbacks of CapsNets. Firstly, since the routing process is excluded, Vote-Attack could mitigate the gradient obfuscation when computing the gradient to generate adversarial samples. Hence, the attack performance of Vote-Attack is expected to be higher than Caps-Attack. Secondly, since the costly routing process is removed from the attack method, Vote-Attack will be more efficient than Caps-Attack.

## 5 EXPERIMENTS

In this section, we verify our proposal via empirical experiments. We first show the effectiveness of Vote-Attack on CapsNets in the regular training scheme and the adversarial training one. We also show the efficiency of Vote-Attack. Besides, we apply Vote-Attack to bypass the recently proposed CapsNet-based adversarial detection method. All the reported scores are averaged over 5 runs.

### 5.1 EFFECTIVENESS OF VOTE ATTACK ON CAPSNETS

*Models:* We take ResNet18 as a CNN baseline. In couter-part CapsNets, we apply resnet18 backbone to extract primary capsules $\boldsymbol{u} \in (64 \times 4 \times 4, 8)$ where the outputs of the backbone are feature maps of the shape $(512, 4, 4)$ and 64 is the number of capsule groups, 8 is the primary capsule size. The primary capsules are transformed to make $64 \times 4 \times 4$ votes $\hat{\boldsymbol{u}} \in (64 \times 4 \times 4, 10, 16)$ with the learned transformation matrices $\boldsymbol{W} \in (64 \times 4 \times 4, 8, 160)$. The size of output capsule is 16, and 10 are the number of output classes. The votes $\hat{\boldsymbol{u}}$ reach an agreement $\boldsymbol{v} \in (10, 16)$ via the dynamic routing meachnism. The length of 10 output capsules are the probabilites of 10 output classes.

*Datasets:* The popular datasets CIFAR10 (Krizhevsky et al., 2009) and SVHN (Netzer et al., 2011) are used in this experiment. The standard preprocess is applied on CIFAR10 for training: 4 pixels are padded on an input of $32 \times 32$, and a $32 \times 32$ crop is randomly sampled from the padded image or its horizontal flip. For $\ell_\infty$-based attacks, the perturbation range is 0.031 (CIFAR10) and 0.047 (SVHN) for pixels ranging in [0, 1]. For $\ell_2$-based attacks, the $\ell_2$ norm of the allowed maximal perturbation is 1.0 for both datasets.

**White-Box Attacks** We train CNNs and CapsNets with the same standard training scheme where the models are trained with a batch size of 256 for 80 epochs using SGD with an initial learning rate of 0.1 and moment 0.9. The learning rate is set to 0.01 from the 50-th epoch. We apply popular $\ell_\infty$-based attacks (FGSM (Goodfellow et al., 2015), BIM (Kurakin et al., 2017), MIM (Dong et al., 2018),PGD (Madry et al., 2018)) and $\ell_2$-based attacks (C&W attack (Carlini & Wagner, 2017b), Deepfool (Moosavi-Dezfooli et al., 2016)) to attack the well-trained models. The hyper-parameters mainly follow the Foolbox tool (Rauber et al., 2017). In CapsNets, Capsules and Votes are taken as targets to attack, respectively.

Table 1: The robust accuracy of ResNets and CapsNets are shown under popular attacks on CIFAR10 and SVHN datasets. Vote-Attack is much more effective than Caps-Attack and compatible with different underlying attacks.

| Model | Target | FGSM | BIM | MIM | PGD | Deepfool-$\ell_2$ | C&W-$\ell_2$ |
|---|---|---|---|---|---|---|---|
| On **CIFAR10 Dataset**, the model accuracy are ResNet 92.18$_{(\pm 0.57)}$ and CapsNet 92.80$_{(\pm 0.14)}$. | | | | | | | |
| ResNet | Logits | 16.6$_{(\pm 0.76)}$ | 0.15$_{(\pm 0.05)}$ | 0$_{(\pm 0)}$ | 0$_{(\pm 0)}$ | 0.08$_{(\pm 0.05)}$ | 0.24$_{(\pm 0.14)}$ |
| CapsNet | Caps | 44.55$_{(\pm 1.6)}$ | 24.43$_{(\pm 1.95)}$ | 21.69$_{(\pm 2.52)}$ | 17.3$_{(\pm 1.35)}$ | 26.55$_{(\pm 0.43)}$ | 18.91$_{(\pm 1.5)}$ |
| | **Votes** | **26.21**$_{(\pm 1.66)}$ | **8.12**$_{(\pm 0.13)}$ | **9.20**$_{(\pm 3.44)}$ | **4.83**$_{(\pm 0.05)}$ | **20.83**$_{(\pm 0.78)}$ | **6.66**$_{(\pm 0.32)}$ |
| On **SVHN Dataset**, the model accuracy are ResNet 94.46$_{(\pm 0.14)}$ and CapsNet 94.16$_{(\pm 0.02)}$. | | | | | | | |
| ResNet | Logits | 14.57$_{(\pm 2.73)}$ | 2.9$_{(\pm 0.47)}$ | 0.06$_{(\pm 0.02)}$ | 0.06$_{(\pm 0.02)}$ | 3.05$_{(\pm 0.45)}$ | 2.16$_{(\pm 0.1)}$ |
| CapsNet | Caps | 58.32$_{(\pm 1.34)}$ | 50.25$_{(\pm 0.88)}$ | 40.09$_{(\pm 1.65)}$ | 34.82$_{(\pm 2.11)}$ | 45.76$_{(\pm 1.17)}$ | 44.29$_{(\pm 1.07)}$ |
| | **Votes** | **49.16**$_{(\pm 1.0)}$ | **31.46**$_{(\pm 0.22)}$ | **14.22**$_{(\pm 0.23)}$ | **8.11**$_{(\pm 0.3)}$ | **39.31**$_{(\pm 0.56)}$ | **27.94**$_{(\pm 0.14)}$ |

The standard test accuracy and the robust accuracy under different attacks are reported in Table 1. The CapsNets and the counter-part CNNs achieve similar performance on normal test data. The strong attack PGD can mislead all the classifications of ResNet. However, it is less effective to attack output capsules. Our Vote-Attack can reduce the robust accuracy of CapsNets significantly across different attack methods. We also check the $\ell_0, \ell_1, \ell_2$ norms of the perturbations created by different attack methods in Appendix B. In most cases, the different norms of perturbations corresponding to Vote-attack is similar to the ones to Caps-attack.

We also verify the effectiveness of Vote-Attack from other perspectives, such as, the targeted attacks, the transferability of adversarial examples and the adversarial robustness on affine-transformed inputs. The experimental details of the targeted Vote Attack are in the Appendix C. The transferability

of the created adversarial examples is investigated in Appendix D. The adversarial examples created by Vote-attack are more transferable than the ones by Caps-attack.

CapsNets are shown to be robust to input affine transformation (Sabour et al., 2017; Gu & Tresp, 2020a). When inputs are affine transformed, the votes in CapsNets also change correspondingly. We also verify the effectiveness of Vote-Attack in case of affine transformed inputs. We consider two cases: 1) The CapsNet built on standard convolutional layers (Sabour et al., 2017) is trained on MNIST dataset and tested on AffNIST dataset. 2) The CapsNet built on a backbone (i.e. ResNet18) is trained on the standard CIFAR10 training dataset and tested on affine-transformed CIFAR10 test images. In both cases, our Vote-Attack achieves higher attack success rates than Caps-Attack. More details about this experiment can be found in Appendix E. This experiment shows that our Vote-Attack is more effective than Caps-Attack when the inputs are affine-transformed.

Under Vote-Attack, the robust accuracy of CapsNets is still higher than that of counter-part CNNs. However, we did claim CapsNets are more robust for two reasons. 1) CapsNets possess more network parameters due to transformation matrices. 2) The potential attacks can reduce the robust accuracy further. This study demonstrates that the high adversarial robustness of CapsNets can be a fake sense, and we should be careful to draw any conclusion about the robustness of CapsNets.

**Adversarial Training** In this experiment, we verify the effectiveness of Vote-Attack in the context of Adversarial Training. We train models with adversarial examples created by Caps-Attack where PGD with 8 iterations is used. For training a more robust model, we also combine Vote-Attack and Caps-Attack to create adversarial examples where a new loss from the two attacks is used.

The underlying attack method used in this experiment is PGD with 40 iterations. The model performance is reported in Table 2 under different training schemes. We can observe that the Vote-Attack (corresponding to the last column) is more effective than Caps-Attack (corresponding to the second last column) under adversarial training. When we include Vote-Attack to improve the adversarial training (AT v.s. AT +Votes), the robust accuracy of CapsNets is increased under both Caps-Attack and Vote-Attack.

The Vote-Attack only attacks part of the model. During adversarial training, the model can adapt the routing process to circumvent the adversarial perturbations. Therefore, it is not effective to do adversarial training only using Vote-Attack.

Table 2: The robustness of CapsNets with different training schemes on CIFAR10 and SVHN datasets: Vote-Attack is also effective to attack models with adversarial training; It can also be applied to improve adversarial training.

| Dataset | Traning | ResNet | | CapsNet | | |
|---|---|---|---|---|---|---|
| | | $A_{std}$ | Logits | $A_{std}$ | Caps | Votes |
| CIFAR10 | Natural | $92.18_{(\pm0.57)}$ | 0 | $92.8_{(\pm0.14)}$ | $17.30_{(\pm1.35)}$ | $4.83_{(\pm0.05)}$ |
| | AT | $79.45_{(\pm1.27)}$ | $43.91_{(\pm0.62)}$ | $75.0_{(\pm0.04)}$ | $45.49_{(\pm0.78)}$ | $43.65_{(\pm0.85)}$ |
| | **AT + Votes** | - | - | $76.42_{(\pm0.37)}$ | $49.62_{(\pm0.56)}$ | $44.12_{(\pm0.32)}$ |
| SVHN | Natural | $94.46_{(\pm0.14)}$ | $0.06_{(\pm0.02)}$ | $94.16_{(\pm0.02)}$ | $34.82_{(\pm2.11)}$ | $8.11_{(\pm0.30)}$ |
| | AT | $87.9_{(\pm0.08)}$ | $36.05_{(\pm0.33)}$ | $86.0_{(\pm0.80)}$ | $33.40_{(\pm1.36)}$ | $30.44_{(\pm1.08)}$ |
| | **AT + Votes** | - | - | $83.89_{(\pm0.73)}$ | $39.13_{(\pm0.96)}$ | $34.92_{(\pm0.98)}$ |

## 5.2 EFFICIENCY OF VOTE ATTACK ON CAPSNETS

In the last subsection, we demonstrate the effectiveness of Vote-Attack from different perspectives. We now show the efficiency of Vote-Attack. In our Vote-Attack, no routing process is involved in both forward inferences and gradient backpropagations. To show the efficiency of Vote-Attack empirically, we record the time required by each attack to create a single adversarial example and average them across the CIFAR10 test dataset. A single Nvidia V100 GPU is used.

The required time is reported in Table 3. The time on SVHN dataset is almost the same as in CIFAR10 since both input space dimensions are the same (*i.e.,* 32, 32, 3). The column corresponding to $A_{std}$ shows the time required to classify a single input image. Compared to the logit attack in CNNs, Caps-Attack in CapsNets requires more time to create adversarial examples since the dy-

namic routing is computationally expensive. Our Vote-Attack can create adversarial images without using the routing part, and reduce the required time significantly. However, the required time is still more than that on CNNs. The reason behind this is that the current deep learning framework is highly optimized on the convolutional operations, less on the voting process.

Table 3: The averaged time required by each attack to create an adversarial example is reported on CIFAR10 test dataset. Vote-Attack requiring less time is more efficient than Caps-Attacks.

| Model | Target | $A_{std}$ | FGSM | BIM | MIM | PGD | Deepfool | C&W |
|-------|--------|-----------|------|-----|-----|-----|----------|-----|
| ResNet | Logits | $4.14ms$ | $12.13ms$ | $83.34ms$ | $165.76ms$ | $324.53ms$ | $186.77ms$ | $409.38ms$ |
| CapsNet | Caps | $5.65ms$ | $17.45ms$ | $120.75ms$ | $242.97ms$ | $471.81ms$ | $607.84ms$ | $612.79ms$ |
|  | **Votes** |  | $\mathbf{14.89}ms$ | $\mathbf{105.09}ms$ | $\mathbf{196.11}ms$ | $\mathbf{414.58}ms$ | $\mathbf{295.28}ms$ | $\mathbf{448.31}ms$ |

## 5.3 Bypassing Class-conditional Capsule Reconstruction based Detection

In this experiment, we demonstrate that class-conditional capsule reconstruction based detection can be bypassed by integrating our Vote-Attack in the detection-aware attack method. Following the work (Qin et al., 2020), we use the original CpasNet architecture (Sabour et al., 2017) for this experiment. The architecture details are shown as follows.

**CapsNets**, two standard convolutional layers, Conv1(C256, K9, S1), Conv2(C256, K9, S2), are used to extract primary capsules of shape (32×6×6, 8). The output capsule of shape (10, 16) can be obtained after the dynamic routing process. The output capsules will be taken as input for a reconstruction net with (FC160-FC512-FC1024-FC28×28). In the reconstruction process, only one of the output capsules is activated, others are masked with zeros. Since the input contains the class information, the reconstruction is class-conditional. The capsules corresponding to the ground-truth class will be activated during training, while the winning capsule (the one with maximal length) will be activated in the test phase.

Two CNN baseline models are considered. **CNN+CR** uses the same architecture without routing and group 160 activations into 10 groups where the sum of 16 activations is taken as a logit. The same class-conditional reconstruction mechanism is used. **CNN+R** does not group 160 activations and reconstructs the input from activations without a masking mechanism. More details of the baseline models can be found in (Qin et al., 2020).

Given an input, it will be flagged as adversarial examples if its reconstruction error is bigger than a given threshold $d(\boldsymbol{x}, \hat{\boldsymbol{x}}) > \theta$. Following (Qin et al., 2020), we set $\theta$ as 95th percentile of reconstruction errors of benign validation images, namely, 5% False postive rate. We report **Success Rate** $S = \frac{1}{K} \sum_{i}^{N} (f(\boldsymbol{x} + \boldsymbol{\delta}) \neq y)$ and **Undetected Rate** $R = \frac{1}{K} \sum_{i}^{N} (f(\boldsymbol{x} + \boldsymbol{\delta}) \neq y) \cap (d(\boldsymbol{x}, \hat{\boldsymbol{x}}) \leq \theta)$. Both detection-agnostic and detection-aware attacks introduced in Sec. 2 are considered.

Table 4: Different attacks are applied to circumvent the class-conditional reconstruction adversarial detection method on FMNIST dataset. The attack success rate and undetected rate ($S/R$) are reported for each attack. The integration of Vote-Attack in the detection-aware attack increases both the attack success rate and the undetected rate significantly.

| Attacks | Model | Target | $A_{std}$ | FGSM | BIM | PGD | C&W |
|---------|-------|--------|-----------|------|-----|-----|-----|
| Detection-agnostic Attack | CNN+R | Logits | 90.95 | 85.8/63.3 | 100/80.0 | 100/75,7 | 86,4/68.8 |
|  | CNN+CR | Logits | 91.79 | 89.4/66.4 | 97.4/70.4 | 97.9/67.9 | 77.3/77.1 |
|  | CapsNet | Caps | 91.85 | 40.2/29.3 | 88.8/53.1 | 90.6/51.4 | 70.7/54.1 |
|  |  | **Votes** |  | **74.8**/46.1 | **94.6**/59.2 | **94.7**/55.3 | **90.5**/50.1 |
| Detection-aware Attack | CNN+R | Logits | 90.95 | 85.3/77.3 | 99.7/95.0 | 100/92.1 | - |
|  | CNN+CR | Logits | 91.79 | 89.3/75.9 | 96.3/82.3 | 96.2/81.2 | - |
|  | CapsNet | Caps | 91.85 | 41.8/37.2 | 87.9/78.7 | 89.7/78.2 | - |
|  |  | **Votes** |  | **76.8/66.5** | **95.1/85.2** | **95.6/86.1** | - |

The results on FMNIST dataset are reported in Table 4. In detection-agnostic attacks, we apply our Vote-Attack to attack CapsNets directly without considering the detection mechanism. The CapsNet

| Clean | 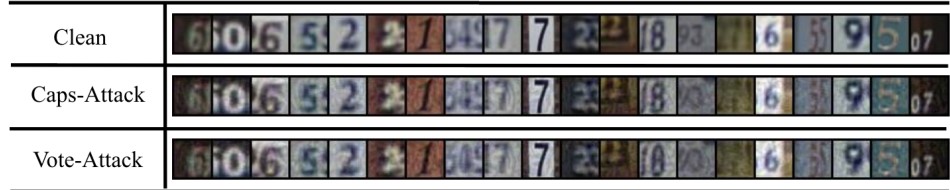 |
|---|---|
| Caps-Attack | |
| Vote-Attack | |

Figure 3: This figure shows the clean images and the corresponding adversarial images created by Caps-Attack and Vote-Attack in a targeted setting. The attack target class is set to the digit 0. The adversarial images created by the two attack methods are visually similar. The observation also echoes the previous findings in Appendix B, where we show that the perturbations created by Caps-Attack and Vote-Attack have similar norms.

used in this experiment is built on standard convolutional layers instead of backbones in previous experiments. Our Vote-Attack still achieve a higher success rate than Caps-Attack. It indicates that the Vote-Attack is effective across different architectures. Furthermore, the undetected rate is also increased correspondingly. In detection-aware attacks, the integration of our Vote-Attack increases the attack success rate and undetected rate significantly. More results on MNIST and SVHN datasets are shown in Appendix F.

Under the class-conditional capsule reconstruction based detection, some of the undetected examples are not imperceptible anymore, as shown in (Qin et al., 2020). Some images are flipped into the attack target classes when attacked, although a small perturbation threshold is applied. Some images are hard to flip, e.g., the ones with a big digit or thin strokes. We also visualize the adversarial examples created by Caps-Attack and our Vote-Attack in Figure 3. More figures and details are shown in Appendix G. We find that there is no obvious visual difference between the adversarial examples created by the two attacks. This finding echoes a previous experiment, where we compute the different norms (i.e., the $\ell_0, \ell_1, \ell_2$ norms) of the created perturbations. The perturbations have similar norms (see Appendix B). Hence, the adversarial examples created by the two attacks are visually similar.

## 6 CONCLUSIONS AND FUTURE WORK

We dive into the inner working of CapsNets and show how it is affected by adversarial examples. Our investigation reveals that adversarial examples can mislead CapsNets by manipulating the votes. Based on the investigation analysis, we propose an effective and efficient Vote-Attack to attack CapsNets. The Vote-Attack is more effective and efficient than Caps-Attack in both standard training and adversarial training settings. Furthermore, Vote-Attack also demonstrates the superiority in terms of the transferability of adversarial examples as well as the adversarial robustness on affine-transformed data. Last but not least, we apply our Vote-Attack to increase the undetected rate significantly of the class-conditional capsule reconstruction based adversarial detection.

The idea of attacking votes of CapsNet can also be applied to different versions of CapsNets. However, some adaptions are required since different CapsNet versions can have significantly different architectures. For instance, in EM-CapsNet (Hinton et al., 2018), a capsule corresponding to an entity are represented by a matrix, and the confidence of the entity's existence is represented by the activation of a single neuron. The possible adaption could be attacking votes by flipping the neuron activations that represents the existence of entities. Recently, many capsule networks have been proposed, to name a few (Hinton et al., 2018; Zhang et al., 2018; Rawlinson et al., 2018; Hahn et al., 2019; Ahmed & Torresani, 2019; Gu & Tresp, 2020a; Tsai et al., 2020; Ribeiro et al., 2020). We leave the further exploration on different versions of CapsNet in future work.

Even though CapsNets still seem to be more robust than counter-part CNNs under our stronger Vote-Attack, it is too early to draw such a conclusion. We conjecture that the robust accuracy of CapsNets can be reduced further. In future work, we will explore more strong attacks as well as the certifications to compare the robustness of CNNs and CapsNets.

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

## A  Two Variants of Vote Attack

We have another two choices when attacking votes in CapsNet directly. **Choices 1**: In Equation (8), we first average the votes and squash the averaged vote. Another choice is to first squash their votes and then average the squashed votes. Our experiments show that this option is similarly effective.

$$\boldsymbol{\delta}^* = \arg\max_{\boldsymbol{\delta} \in \nabla} \mathcal{H}(\log(\frac{1}{N}\sum_{i=1}^{N} g(f_v^i(\boldsymbol{x}+\boldsymbol{\delta}))), \boldsymbol{y}). \tag{9}$$

**Choices 2**: Another choice is to average the loss caused by all votes. Instead of opimizing on the loss computed on the squahed averaged vote, we can compute the loss of individual vote seperatedly and average them, namely,

$$\boldsymbol{\delta}^* = \arg\max_{\boldsymbol{\delta} \in \nabla} \frac{1}{N}\sum_{i=1}^{N} \mathcal{L}(g(f_v^i(\boldsymbol{x}+\boldsymbol{\delta})), \boldsymbol{y}). \tag{10}$$

The loss of each vote can differ from each other significantly. The large part of loss can be caused by a small part of votes. In other words, the gradients of received by the input can be caused mainly by a few too strong votes. This choice is less effective, compared to the one in Equation (8).

We use the same emperimental setting as in Sec. 5. Under the same PGD attack on CIFAR10 dataset, the robust accuracy corresponding to the choice 1 is $4.06_{(\pm 1.12)}$, and it is effective, similar to Equation (8). The choice 2 with the robust accuracy $43.31_{(\pm 2.46)}$ does not work well since the gradients received by inputs are dominated only by a small part of votes.

## B  Norms of Perturbations Created by Different Attacks

On CIFAR10 and SVHN datasets, we compute the different norms of perturbations created by different attacks. On each dataset, we first select the examples that are successfully attacked by both Vote-Attack and Caps-Attack on CapsNets as well as the corresponding attack on ResNets from the test dataset. Then, we obtain the created perturbations created by the corresponding attacks. The $\ell_0$, $\ell_1$ and $\ell_2$ norm of perturbations are shown in Table 5 on CIFAR10 dataset and Table 6 on SVHN dataset.

In most cases, Vote-Attack and Caps-Attack create perturbations with similar norms. Under BIM attack, we can observe that $\ell_1$ and $\ell_2$ norms corresponding to Vote-Attack are higher than the ones to Caps-Attack. Both are smaller than the ones corresponding to other multi-step attacks (**e.g.,** PGD). The reason behind this is that the BIM attack does not converge since only 10 iterations are used by default in FoolBox tool (50 iterations in PGD). Given the same iterations before the convergence, Vote-Attack accumulates the relatively consistent gradients. Vote-Attack converges faster than Caps-Attack, which explains our observation.

In addition, the $\ell_2$ attack find the minimal perturbations to misled the classifier. The different norms of perturbations is small. Our Vote-Attack finds samller perturbations in SVHN dataset and similar ones in CIFAR10 dataset. This obervation indicate that the performance of attack method can also depend on the datasets.

## C  Vote Targeted Attack

We create adversarial examples in targeted attack settings on CIFAR10 and SVHN datasets. The used models are the same as in the untargeted setting. The target classes are selected uniformly at random from the non-ground-truth classes. The attack is successful if the created adversarial examples are classified as the corresponding target classes by the underlying classifier.

The attack success rate (%) is reported in Table 7. In the targeted attack setting, our Vote-Attack achieves a significantly higher attack success rate than Caps-Attack. This exeriment show that our Vote-Attack is still effective when extended to the targeted attack setting.

Table 5: The $\ell_0$, $\ell_1$, and $\ell_2$ norms of perturbations created by different attacks are shown on CIFAR10 dataset. Overall, the perturbations created by our Vote-Attack have similar norms to the ones by Caps-Attack.

| | Model | Target | FGSM | BIM | MIM | PGD | Deepfool-$\ell_2$ | C&W-$\ell_2$ |
|---|---|---|---|---|---|---|---|---|
| $\ell_0$ **norm** | ResNet | Logits | 3054.7 | 2797.1 | 3071.9 | 3057.4 | 1533.5 | 2942.2 |
| | CapsNet | Caps | 3054.5 | 2489.2 | 3071.6 | 3066.3 | 1431.7 | 2977.4 |
| | | **Votes** | 3054.6 | 2741.1 | 2523.9 | 3065.7 | 1534.8 | 2978.1 |
| $\ell_1$ **norm** | ResNet | Logits | 93.96 | 53.07 | 77.89 | 77.38 | 0.21 | 0.51 |
| | CapsNet | Caps | 93.93 | 28.79 | 78.86 | 53.36 | 0.32 | 0.42 |
| | | **Votes** | 93.91 | 43.68 | 78.71 | 54.03 | 0.32 | 0.51 |
| $\ell_2$ **norm** | ResNet | Logits | 1.7041 | 1.1089 | 1.4974 | 1.4849 | 0.0059 | 0.0105 |
| | CapsNet | Caps | 1.7037 | 0.6471 | 1.5066 | 1.1035 | 0.0092 | 0.0087 |
| | | **Votes** | 1.7035 | 0.6753 | 1.5047 | 1.1155 | 0.0091 | 0.0104 |

Table 6: The $\ell_0$, $\ell_1$, and $\ell_2$ norms of perturbations created by different attacks are shown on SVHN dataset. In $\ell_\infty$-attack methods, the perturbations created by our Vote-Attack have similar norms to the ones by Caps-Attack. In $\ell_2$-attack methods, our Vote-attack can find smaller perturbations to fool the underlying classifier.

| | Model | Target | FGSM | BIM | MIM | PGD | Deepfool-$\ell_2$ | C&W-$\ell_2$ |
|---|---|---|---|---|---|---|---|---|
| $\ell_0$ **norm** | ResNet | Logits | 3066.9 | 2854.1 | 3071.9 | 3066.0 | 1754.4 | 2972.7 |
| | CapsNet | Caps | 3067.4 | 2552.2 | 3071.8 | 3070.3 | 2103.4 | 2931.1 |
| | | **Votes** | 3067.2 | 2587.6 | 3071.8 | 3069.7 | 875.0 | 2924.9 |
| $\ell_1$ **norm** | ResNet | Logits | 94.83 | 59.18 | 80.75 | 111.37 | 0.48 | 0.65 |
| | CapsNet | Caps | 94.88 | 32.99 | 77.77 | 79.37 | 0.52 | 0.87 |
| | | **Votes** | 94.86 | 35.06 | 78.00 | 80.33 | 0.24 | 0.15 |
| $\ell_2$ **norm** | ResNet | Logits | 1.7136 | 1.2041 | 1.5357 | 2.1657 | 0.0141 | 0.0149 |
| | CapsNet | Caps | 1.7142 | 0.7283 | 1.4920 | 1.6464 | 0.0161 | 0.0172 |
| | | **Votes** | 1.7140 | 0.7677 | 1.4954 | 1.6442 | 0.0074 | 0.0029 |

# D    TRANSFERABILITY OF ADVERSARIAL EXAMPLES

We also investigate the transferability of adversarial examples created by Caps-Attack and Vote-Attack on CIFAR10 dataset. We consider three models, VGG19 (Simonyan & Zisserman, 2015), ResNet18 and CapsNets. The PGD is used as the underlying attack. We measure the transferability using Transfer Sucess Rate (TSR).

The TSR of different adversarial examples is reported in Table 8. The adversarial examples created on CNNs are more transferable. Especially, the ones created on ResNet18 can be transferred to CapsNets very well. The reason behind this is that CapsNets also the ResNet18 bone to extract primary capsules. By comparing the last two columns in Table 8, we can observe that the adversarial example created by Vote-Attack is more transferable than the ones created by Caps-Attack.

# E    ADVERSARIAL ROBUSTNESS ON AFFINE-TRANSFORMED DATA

CapsNets learn equivariant visual representations. When inputs are affine transformed, the votes also changes correspondingly. In this experiment, we aim to verify the effectiveness of Vote-Attack when inputs and their votes in Capsnets changed. The model is trained the same as before. We translate the test images with 2 pixels randomly and rotate the images within a given pre-defined degree.

Table 7: The targeted attack success rates (%) are shown on CIFAR10 and SVHN datasets. In the targeted attack setting, our Vote-Attack is significantly more effective than Caps-Attack when combined with popular attacks.

| Model | Target | FGSM | BIM | MIM | PGD | Deepfool-$\ell_2$ | C&W-$\ell_2$ |
|---|---|---|---|---|---|---|---|
| | | On **CIFAR10 Dataset**, the model accuracy are ResNet 92.18$_{(\pm0.57)}$ and CapsNet 92.80$_{(\pm0.14)}$. | | | | | |
| ResNet | Logits | 39.13$_{(\pm3.11)}$ | 98.47$_{(\pm0.68)}$ | 99.71$_{(\pm0.33)}$ | 99.97$_{(\pm0.04)}$ | 10.47$_{(\pm0.11)}$ | 97.99$_{(\pm1.39)}$ |
| CapsNet | Caps | 9.58$_{(\pm0.16)}$ | 27.91$_{(\pm2.11)}$ | 48.38$_{(\pm0.21)}$ | 65.94$_{(\pm0.92)}$ | 9.43$_{(\pm0.48)}$ | 34.07$_{(\pm1.38)}$ |
| | **Votes** | **10.67**$_{(\pm0.32)}$ | **32.66**$_{(\pm2.09)}$ | **61.08**$_{(\pm4.71)}$ | **75.35**$_{(\pm0.91)}$ | **9.55**$_{(\pm0.64)}$ | **41.41**$_{(\pm5.85)}$ |
| | | On **SVHN Dataset**, the model accuracy are ResNet 94.46$_{(\pm0.14)}$ and CapsNet 94.16$_{(\pm0.02)}$. | | | | | |
| ResNet | Logits | 43.06$_{(\pm3.37)}$ | 91.72$_{(\pm0.42)}$ | 98.15$_{(\pm0.02)}$ | 99.78$_{(\pm0.04)}$ | 11.84$_{(\pm0.45)}$ | 93.97$_{(\pm0.82)}$ |
| CapsNet | Caps | 5.82$_{(\pm0.06)}$ | 38.58$_{(\pm0.59)}$ | 49.04$_{(\pm0.89)}$ | 68.94$_{(\pm2.11)}$ | 6.82$_{(\pm1.12)}$ | 44.64$_{(\pm0.96)}$ |
| | **Votes** | **7.28**$_{(\pm1.73)}$ | **48.25**$_{(\pm1.02)}$ | **65.35**$_{(\pm0.28)}$ | **91.68**$_{(\pm1.06)}$ | **7.57**$_{(\pm1.06)}$ | **62.93**$_{(\pm0.55)}$ |

Table 8: The transferability of adversarial examples created on CNNs and CapsNets on CIFAR10 dataset: the ones created on CNNs are more transferable than on CapsNets; the ones created with Vote-Attack are more transferable than the ones with Caps-Attack.

| | | Attacks on Source Model | | | |
|---|---|---|---|---|---|
| | | VGG19 (Logits) | ResNet18 (Logits) | CapsNet (Caps) | CapsNet (Votes) |
| Target Models | VGG19 | 83.79$_{(\pm0.18)}$ | 93.94$_{(\pm0.28)}$ | 35.64$_{(\pm0.96)}$ | 41.49$_{(\pm0.19)}$ |
| | ResNet18 | 71.81$_{(\pm1.04)}$ | 97.26$_{(\pm1.84)}$ | 37.59$_{(\pm6.25)}$ | 43.45$_{(\pm8.13)}$ |
| | CapsNet | 80.38$_{(\pm1.79)}$ | 97.53$_{(\pm0.57)}$ | 46.43$_{(\pm5.56)}$ | 55.34$_{(\pm6.26)}$ |

The robust accuracy of affine-transformed images is shown in Table 9 on CIFAR10 dataset. Under different rotation degrees, our Vote-Attack is still effective. It consistently reduces the robust accuracy of CapsNets, when compared to Caps-Attack.

Table 9: When inputs are affine-transformed in CIFAR10 dataset, the Vote-Attack is still more effective to create adversarial examples than Caps-Attack.

| Model | Target | $(0, \pm0°)$ | $(\pm2, \pm15°)$ | $(\pm2, \pm30°)$ | $(\pm2, \pm60°)$ | $(\pm2, \pm90°)$ |
|---|---|---|---|---|---|---|
| ResNet | $A_{std}$ | 92.18$_{(\pm0.57)}$ | 85.64$_{(\pm0.46)}$ | 68.11$_{(\pm1.12)}$ | 48.47$_{(\pm0.30)}$ | 42.07$_{(\pm0.22)}$ |
| | Logits | 0 | 0 | 0 | 0 | 0 |
| CapsNet | $A_{std}$ | 92.8$_{(\pm0.14)}$ | 86.09$_{(\pm0.39)}$ | 69.44$_{(\pm1.96)}$ | 49.37$_{(\pm2.43)}$ | 42.62$_{(\pm1.64)}$ |
| | Caps | 17.3$_{(\pm1.35)}$ | 5.82$_{(\pm1.86)}$ | 2.89$_{(\pm1.05)}$ | 1.63$_{(\pm0.51)}$ | 1.11$_{(\pm0.38)}$ |
| | **Votes** | **4.83**$_{(\pm0.05)}$ | **1.15**$_{(\pm0.38)}$ | **0.54**$_{(\pm0.22)}$ | **0.32**$_{(\pm0.16)}$ | **0.23**$_{(\pm0.08)}$ |

We also conduct experiments on AffNIST dataset. In this experiment, the original CapsNet architecture and the original CNN baseline in (Sabour et al., 2017) are used. The modes are trained on standard MNIST dataset and tested on AffNIST dataset. In AffNIST dataset, the MNIST images are transformed, namely, rotated, translated, scaled, or sheared. More details about this dataset are in this resource [1]. The perturbation threshold and the attack step size are set to 0.3 and 0.01, respectively. The other hyper-parameters are defaults in the Foolbox tool (Rauber et al., 2017).

The test accuracy on the untransformed test dataset ($A_{std}$), the accuracy on the transformed dataset ($A_{aff}$) and the robust accuracy under different attacks are reported in Table 10. Our Vote-Attack achieve higher attack success rates than Caps-Attack.

---

[1]https://www.cs.toronto.edu/ tijmen/affNIST/

Table 10: The test accuracy on the dataset with untransformed images and the one on the dataset with transformed images are reported (in %). CapsNet achieves better transformtation robustness than the original CNN baseline. The robust accuracy of different models are also reported under different attacks. We can observe that it is more effective to attack Votes instead of output capsules in CapsNet.

| Model | Target | $A_{std}$ | $A_{aff}$ | FGSM | BIM | MIM | PGD |
|-------|--------|-----------|-----------|------|-----|-----|-----|
| ResNet | Logits | 99.22 | 66.08 | 10.18 | 0 | 0 | 0 |
| CapsNet | Caps | 99.22 | 79.12 | 15.61 | 4.27 | 1.01 | 0.48 |
|  | **Votes** |  |  | 10.43 | 1.33 | 0 | 0 |

## F BYPASSING CLASS-CONDITIONAL RECONSTRUCTION ON MNIST, FMNIST AND SVHN

The integration of our Vote-attack into detection-aware attack is effective to bypass the class-conditional reconstruction detection method. To verify this, we also conduct experiments on different datasets, such as MNIST and SVHN. The results are reported in Table 11. On the All three datasets, both detection-aware and detection-agnostic attacks achieve high attack success rate and undetected rate, when combined with our Vote-attack.

Table 11: Different attacks are applied to circumvent the class-conditional reconstruction adversarial detection method. The attack success rate and undetected rate ($S/R$) are reported for each attack. On all the three popular datasets, the integration of Vote-Attack in the detection-aware attack increases both the attack success rate and the undetected rate significantly.

| DataSet | Model | $A_{std}$ | Attacks | Target | FGSM | BIM | PGD |
|---------|-------|-----------|---------|--------|------|-----|-----|
| **MNIST** | CapsNet | 99.41 | Detection-agnostic | Caps | 13.3/6.3 | 73.3/31.7 | 77.9/33.1 |
|  |  |  |  | Votes | 38.8/15.1 | 92.3/35.4 | 93.4/34.1 |
|  |  |  | Detection-aware | Caps | 16.1/13.6 | 71.7/52.7 | 77.2/57.8 |
|  |  |  |  | Votes | 44.8/34.9 | 92.1/66.8 | 93.4/67.1 |
| **FMNIST** | CapsNet | 91.85 | Detection-agnostic | Caps | 40.2/29.3 | 88.8/53.1 | 90.6/51.4 |
|  |  |  |  | Votes | 74.8/46.1 | 94.6/59.2 | 94.7/55.3 |
|  |  |  | Detection-aware | Caps | 41.8/37.2 | 87.9/78.7 | 89.7/78.2 |
|  |  |  |  | Votes | 76.8/66.5 | 95.1/85.2 | 95.6/86.1 |
| **SVHN** | CapsNet | 91.32 | Detection-agnostic | Caps | 83.2/78.1 | 99.1/92.3 | 99.6/92.2 |
|  |  |  |  | Votes | 95.5/88.8 | 99.9/93.2 | 99.9/93.3 |
|  |  |  | Detection-aware | Caps | 84.2/80.1 | 97.8/95 | 97.8/94.7 |
|  |  |  |  | Votes | 90.6/90.8 | 100/96.7 | 100/96.8 |

## G VISUALIZING UNDETECTED ADVERSARIAL EXAMPLES

We also visualize the adversarial examples created by Caps-Attack and Vote-Attack in Figure 4. In this experiment, following (Qin et al., 2020), we use a detection-aware attack method and set the attack target class is 0. The standard setting 0.047 is used in the case of input range [0, 1], which corresponds to 12 of the pixel range of 255. In Figure 4, Some adversarial examples are flipped to target class to human perception, although the perturbation threshold is small. For some examples, it is hard to flip them, e.g., the ones with a big digit and thin strokes.

By comparing the adversarial examples created by Caps-Attack and Vote-Attack, we can find that there is no obvious visual difference between the adversarial examples. The observation also echos with our experiment in Appendix B. In that experiment, we compute the different norms of the perturbations created by different methods. The results in Table 5 and 6 show the perturbations created by Caps-Attack and Vote-Attack have similar norms. Hence, the adversarial examples created by Caps-Attack and Vote-Attack are also visually similar.

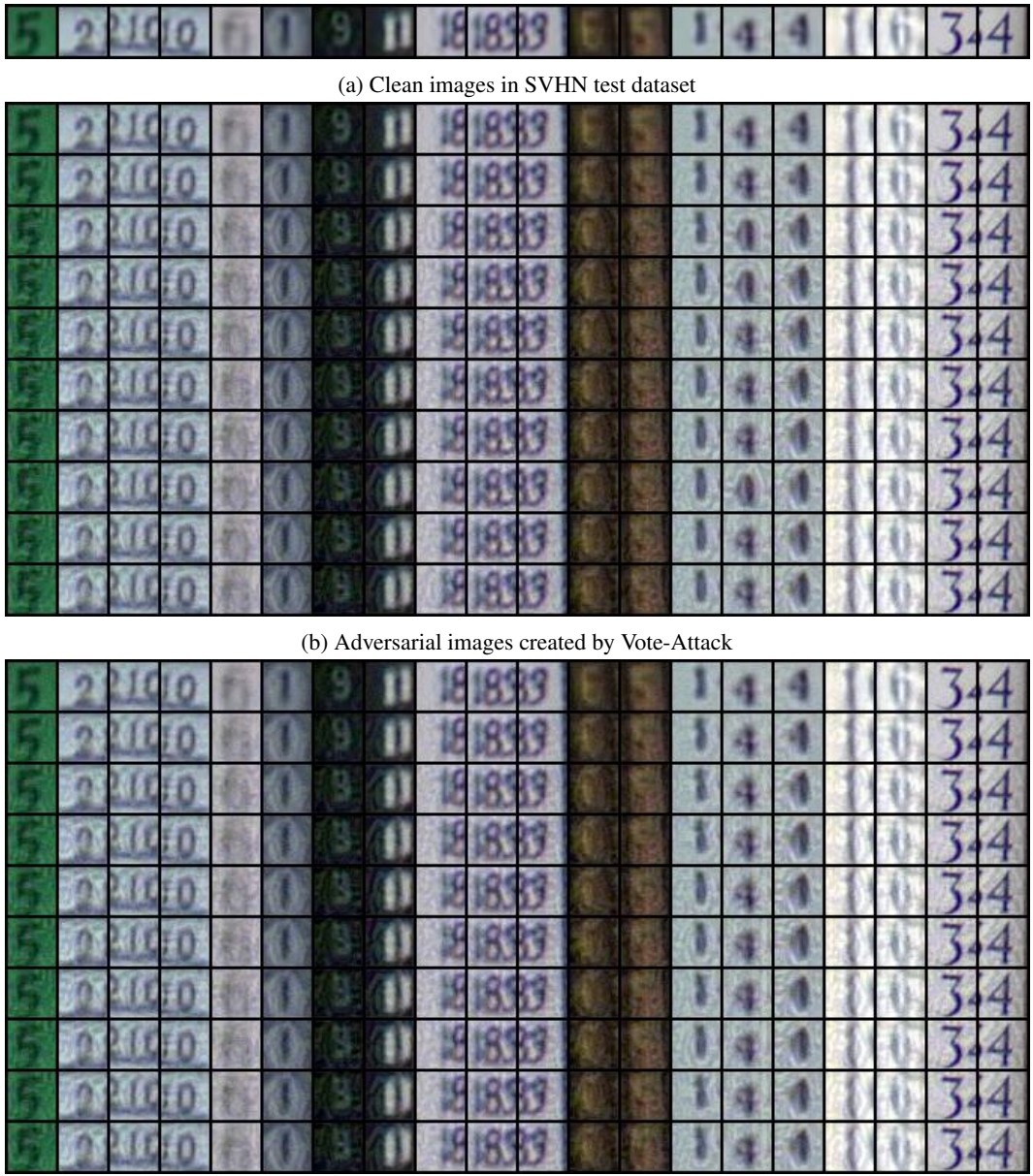

(a) Clean images in SVHN test dataset

(b) Adversarial images created by Vote-Attack

(c) Adversarial images created by Vote-Attack

Figure 4: The first subfigure shows clean test images of the SVHN dataset. The second subfigure shows the adversarial images created by Caps-Attack. Different rows correspond to different weights to reduce reconstruction error in Equation (6) (i.e., the second attack step in detection-aware attack method). Some images are flipped, and some hard ones are not. The images in the third subfigure are the adversarial images created by Vote-Attack. There is no obvious visual difference between the adversarial examples created by the two attacks. To be noted that the images are randomly selected (not cherry picked).

