# OpenReview forum: "Effective and Efficient Vote Attack on Capsule Networks"
_ICLR.cc/2021/Conference — ICLR 2021 Poster_

### Official Review · AnonReviewer3 · 2020-10-25
**An adversarial attack designed for Capsule Network**

**Rating:** 6
**Confidence:** 3

**Review:**

Authors argue that in order to attack Capsule network more effectively one should consider their inner workings, i.e. iterative routing. They propose two reasons behind the relative robustness of capsnets vs cnns: 1) gradient obfuscating 2) being more computationally intensive. Therefore, they propose a new attack which attacks a CapsNet running only 1 routing iteration.
The attacks are shown to be more effective than targeting the 3-iteration capsnets directly. Which suggests the gradients for 1-iteration CapsNets are close enough to a 3-iteration version that the adversarial images fool a 3-iteration capsnet too.

They also show that their attacks stay undetected by Qin et al method (class conditional reconstruction). However, the point of Qin et al was to some degree that although adversaries exist that Qin et al method will not detect, such adversaries have semantical resemblance to the target class and they are not undetected to human eye. i.e. the change is not scattered noise and is not imperceptible anymore. Therefore, to fully claim that their method fools Qin et al, a set of advarsarial images should be visualized to showcase they being imperceptible changes while going undetected.

Authors also provide a study on how attacking a capsule network changes the voting/routing mechanism. They find that attackers make the votes orthogonal to the target class parameters and therefore, remove votes from contributing.
I was not able to see the connection of why this finding leads to their advarsarial attack. So maybe more clarification is required.

But given gradient obfuscating as a reason for capsnet robustness their attack is intuitive (rather than attacking the 3 iterations attack a 1 iteration version).

The study by itself is interesting and can lead to further robustness of CapsuleNetworks.

One caveat of the vote attack is that it seems it is most effective where there is only one Capsule layer. It seems that adding more and more capsule layers would make the model further robust to adversarial attacks. Hinton et al 2018 specifically had 3 capsule layers. An study with several capsule layers would enhance this papers argument and showcase its effectiveness better.

Pros: The paper is relatively well written. Authors explain their attack and motivate their research well. Also the experimental setup covers several attacks. There is novelty both in the idea and in the voting study.

Cons: Not covering multi capsule layer architectures, weak argument against Qin et al (not visualizing their detection aware advarsaries). Also, given that Michels et al 2019 has shown CapsNets are not robust to all attacks and already other attacks exist that CapsNets fail on, the relative interest domain of this paper is limited to researchers working on Capsule Networks.

--------------------------------------------------POST AUTHOR RESPONSE
1) Thank you for adding Fig. 3. My concern about the attacks being noticeable is resolved.
3 & 4) The scope of this work is still limited to Specifically dynamic routing, which we already know are susceptible to black box attack (the community that already exists on architecture un-aware attacks) and some other attacks. As author's mention it is not trivial to apply their attack to other capsule architectures either.
However, I find their various analysis informative.
Therefore, I am increasing my score to 6.

---

> ### Author Response · Authors · 2020-11-18
> **Response to Reviewer 3**
>
> We appreciate that reviewer recognizes the novelty of our idea and the importance of our study. Our responses to the concerns of the reviewer are as follows:
>
> 1.) In undetected adversarial examples in Qin et al [1], the change is not scattered noise and is not imperceptible anymore
>
> The Qin paper [1] claims that the class-conditional based CapsNet behaves better than counter-part CNNs when detecting adversarial examples, and some undetected adversarial examples are flipped to attack target classes.
>
> In Table 4, we show that, under our Vote-Attacks with a strong underlying attack method, the undetected rate on CapsNets can be very close to that on counter-part CNNs. In the revised version of our paper, we also visualize the adversarial images created by Caps-Attack and our Vote-Attack in Figure 3 as well as in Appendix F.  The figures show the adversarial example created by the two attacks are visually similar. The observation also echoes the findings in Appendix B, where we compute the norms of perturbations created by the two networks, and we find that the perturbations corresponding to them have similar norms.
>
> 2.) Attackers remove votes from contributing by making the votes orthogonal to the target class parameters. Why this finding leads to their adversarial attack?
>
> When a clean image is classified by a well-trained CapsNet, the most votes are positive on the ground-truth class. When the artificial perturbation is created to fool CapsNet, most votes of CapsNets become invalid. This finding indicates that the adversarial examples fool CapsNet by manipulating the votes implicitly. To make the attack more effective, we propose to attack votes explicitly. It is possible to do so since the votes have semantic meaning, which corresponds to output classes.
>
>
> 3.) Multi capsule layer architectures Hinton 2018
>
> The overall idea of directly attacking votes (intermediate parts) is useful to attack different versions of CapsNets. For instance, in EM-CapsNets (Hinton 2018 [2]), one way we can image to attack them is to flip the activation unit from alpha to (1- alpha) in each layer. Some more adaptions might be required to deal with the different architectures.
>
> Please see our general response and our discussion in the last section of the revised submission.
>
>
> 4.) Given that Michels et al 2019 [3], the relative interest domain of this paper is limited to researchers working on Capsule Networks:
>
> Michels et al 2019 only checks the robustness of CapsNet with standard convolutional layers. The SOTA CapsNets are often built on ResNet backbones. We carefully check the robustness of backbone-based CapsNet with popular attack protocol. The experiment results are shown in Table 1. In the table, CapsNet does outperform the counter-part CNNs (i.e., ResNet) under all popular attacks listed in our paper.
>
> The observation above shows that the adversarial robustness of CapsNet is worthy of further investigations. In this work, we focus on attacks on CapsNet. Specially, we propose an effective and efficient Vote Attack based on the architecture of CapsNet.
>
> Last but not least, our study also show that the attacks designed for CNNs do not generalise very well to networks with different architectures. For the researchers that are interested in adversarial robustness of CNNs, this finding might inspire them to develop new attacks that generalize to different CNN architectures.
>
> Hence, our work can be of interest to a broad audience.
>
> [1] Qin, Y., Frosst, N., Sabour, S., Raffel, C., Cottrell, G. and Hinton, G., 2019. Detecting and Diagnosing Adversarial Images with Class-Conditional Capsule Reconstructions. ICLR 2020.
> [2] Hinton, Geoffrey E., Sara Sabour, and Nicholas Frosst. "Matrix capsules with EM routing." International conference on learning representations. 2018.
> [3] Michels, Felix, et al. "On the vulnerability of capsule networks to adversarial attacks." arXiv preprint arXiv:1906.03612 (2019).

---

### Official Review · AnonReviewer2 · 2020-10-28
**A good paper with some areas of improvement.**

**Rating:** 5
**Confidence:** 3

**Review:**

This paper presents a new adversarial attack targeting the votes derived from the primary capsules of a capsule network. It demonstrates that this is a stronger attack against capsules networks than an attack directly optimizing the output logits of a capsule model. This paper has some novel contributions to the literature but can be improved in a few areas.

The paper spends a fair bit of time discussing the efficiency of their attack but does not meaningfully motivate this focus. They claim that the time it takes to compute adversarial attacks against capsule networks may have contributed to the claimed robustness, but all the papers they cite which discuss the adversarial robustness of capsule networks measures the success rate with respects to the number of attack optimization steps as opposed to optimization time. Furthermore as shown in table 3, their new attack does not significantly decrease the attack creation time for most optimization strategies. I believe the focus on attack optimization time should be removed from the paper entirely, and the space can be dedicated to some of the results relegated to the appendix such as black box attacks, or the class conditional reconstruction detection.

By creating an attack that specifically targets the votes of the primary capsules this paper is able to increase the success rate of adversarial attacks against capsule networks. It does this however, by effectively not attacking the capsule part of a capsule network. Rather they have created an attack which optimizes the features extracted by the CNN feature extractor used to derive primary capsules. This is an interesting finding, and illustrates that capsules networks are more vulnerable to adversarial attack than perhaps previously believed due to their reliance on CNN feature extraction, but it is important to note that it does so by optimizing for activations of non-capsule components of a capsule network. This is not a major flaw of the paper, but i believe it warrants some discussion, space providing.

This paper also presents the success rate and undetected rate of their new attack against the class conditional reconstruction attack presented by Yao et al. This section improves the paper, but there are some omissions. Namely they do not in the main text nor in the appendix visualize the resultant attacks. This is an issue as both Yao et al (2020) (Detecting and diagnosing adversarial images with class-conditional capsule reconstructions) and  Yao et al (2020) (Deflecting Adversarial Attacks) show that the undetected attacks often resemble the target class, even under small epsilon bounds. This paper also does not address the additional defense mechanisms presented in Deflecting Adversarial Attacks, which were shown to drastically increase the attack detection rate, specifically in colour datasets such as cifar10.

This paper would be improved by addressing those 3 main issues.

-------------------
smaller issues,

the text describing figure 2 is confusing, and i am not entirely sure what the point of the figure it. perhaps more space could be dedicated the distinguishing the b and c plots and discussing why this motivates the work.

there is a missing equation link in section 4

there is a missing table link in appendix E



----POST AUTHOR RESPONSE --------

1.) Efficiency of our Vote-Attack:

It is clear that the vote attack is more time efficient than other attacks, but there is no clear motivation for this improvement. To my knowledge the attack creation time has never been a barrier to adversarial research, nor has it prevented real world adversarial attacks. As a result this focus of the paper simply confuses the reader, be spending time addressing an issue that is not important in research or practical settings.

2.) Optimizing for activations of non-capsule components of a capsule network:

In the authors response they discuss the semantic meaning of the votes of capsules. This too is a bit of a red herring. Although when discussing the motivation behind capsules, the potential for semantically meaningful capsule votes is invoked, there is nothing in the training procedure that ensures that the activations of the capsules correspond directly to features that humans would find semantically meaningful. In my original review i mentioned that by not attacking the output of the capsules after the routing procedure, this attack was simply optimizing for representations extracted from a standard neural network. In this way this work is similar to the representation attacks first presented by [1] which showed the success of representation attacks on standard neural networks.

3.) The undetected attacks often resemble the target class, under the class-conditional capsule reconstructions detection:

Th addition of the attack visualizations is an improvement but the authors do not specify which attacks are successful and undetected and a few of the visualized attacks do indeed resemble the target class.

4.) The additional defense mechanisms presented in Deflecting Adversarial Attacks:

The authors are right to point out the scope of the paper, and it is perhaps unreasonable to expect this paper to address these defence mechanisms, but their inclusion would greatly strengthen the paper.

[1] Sara Sabour, Yanshuai Cao, Fartash Faghri, and David J Fleet. Adversarial manipulation of deep
representations. In ICLR, 2016.

---

> ### Author Response · Authors · 2020-11-18
> **Response to Reviewer 2**
>
> We thank the reviewer for the thoughtful comments. Our responses to the misunderstandings and the concerns are as follows:
>
> 1.) Efficiency of our Vote-Attack:
> We claim that our Vote-Attack is more efficient than Caps-Attack. Given the same attack optimization step, Vote-Attack takes less time to create an adversarial example since we avoid the expensive routing process by attacking votes directly.
>
> In table 3, our Vote-Attack reduces the adversarial example creation time under Caps-Attack in a range that varies from 12.1% to 51.4%. It makes the attack efficiency closer to that on CNNs. In addition, in CapsNet, the reshaping of feature maps into primary capsules and the transformation process is not optimized by the deep-learning framework. Hence, there is still much time left after we remove routing. Actually, the computational cost of the routing process is high.
>
>
> 2.) Optimizing for activations of non-capsule components of a capsule network:
>
> The feature maps extracted by CNNs are first reshaped into primary capsules. The primary capsules are transformed to make votes. Our Vote-Attack attacks the votes directly. The votes could be seen as deep features to some extent. Different from the features in traditional CNNs, the votes of CapsNet have semantic meanings, which correspond to output classes. Base on the findings, we can attack the votes by formulating a new attack loss. However, the explicit semantic meanings of features in CNNs are unavailable. Hence, our Vote-Attack is designed specifically for CapsNet.
>
>
> 3.) The undetected attacks often resemble the target class, under the class-conditional capsule reconstructions detection:
>
> We visualize the adversarial example created by Caps-Attack and our Vote-Attack in Figure 3 as well as in Appendix F. The figures show the adversarial example created by the two attacks are visually similar. The observation also echoes the findings in Appendix B, where we find that the perturbations created by Caps-Attack and our Vote-Attack have similar norms.
>
> 4.) The additional defense mechanisms presented in Deflecting Adversarial Attacks:
>
> In [1], they focus on deflect adversarial examples. In our case, we mainly study the effectiveness and efficiency of our Vote-Attack. The human perception of adversarial examples as well as the metrics to measure the distance between clean images and their adversarial counterparts are not the focus of this paper. We leave the intensive investigation on this topic with our Vote-Attack in future work.
>
> In addition, other practical reasons stop us from including more detection mechanisms: 1) In the experiments to bypass detection on CapsNets, we mainly follow the experimental setting in work [2]. 2) Due to the limited space, we not able to include more detection methods and the corresponding background knowledge.
>
> [1] Qin, Y., Frosst, N., Sabour, S., Raffel, C., Cottrell, G. and Hinton, G., 2019. Detecting and Diagnosing Adversarial Images with Class-Conditional Capsule Reconstructions. ICLR 2020.
> [2] Qin, Yao, et al. "Deflecting Adversarial Attacks." arXiv preprint arXiv:2002.07405 (2020).

---

### Official Review · AnonReviewer4 · 2020-10-28

**Rating:** 8
**Confidence:** 2

**Review:**

The paper investigates robustness of Capsule Networks (CapsNets) under adversarial attacks and makes several interesting observations around the behaviour of CapsNets under attack regime which have been used later to design a new attack against these types of networks (vote attack).  Through several experiments, they show the proposed vote-attack can reduce the robust accuracy of CapsNets significantly across different methods. Authors analyze the effectiveness of this attack from different perspectives (i.e.  transferability of adversarial examples, the adversarial robustness on affine-transformed inputs).


- Correctness and Clarity: The paper seems correct and concise and they experiment different aspects of the methods for effectiveness. The paper is also in general  well-written and easy to follow.
- Reproducibility: They have also provided several tables and figures on the main paper and appendix to help with understanding of the results and method. They also provided detailed description of the methods, training and experimental setup.  It would be great if authors could also release the code to reproduce the results.
Relation to prior works: I am curious to see if the proposed attack can still be effective for Self-Routing Capsule Networks (Hanh et al.)  and the other family of CapsNets in which the different routing mechanism has been used.
Additional Feedback and Suggestions: There are some editorial issues, like missing ref to equations (e.g. Equation ??) , and missing name tags for numbered items (e.g. Table 4.)

---

> ### Author Response · Authors · 2020-11-18
> **Response to Reviewer 4**
>
> We first thank the reviewer for the positive feedback. Our responses are as follows:
>
> 1.) We will release our code in the future.
>
> 2.) In the revised version of our paper, we discuss the generalization of our vote attack to different CapsNet versions in the last section. A similar discussion can also be found in our general response.
>
> 3.) Thanks for pointing out the editorial issues. We fixed them in our new version.

---

### Official Review · AnonReviewer1 · 2020-10-28
**Good submission on attacking CapsNets in the voting process**

**Rating:** 6
**Confidence:** 4

**Review:**

I think the approach proposed by the authors to attack CapsNets and show how they are affected by adversarial examples is solid. It is backed by a comprehensive experimental design and empirical evidence, demonstrating that attacking the voting process of routing is better than attacking output capsules directly.
Comments:
-- I am not sure that this statement is accurate "Since the routing process is the main difference between CapsNets and CNNs,...."....Capsules are built under different premises one of which is the routing process.
--   "In couter-part[sic] CapsNets, we apply ResNet18 backbone to extract primary capsules".......What's the point of using a ResNet-18 backbone with CapsNets when the whole point of CapsNets is to adhere to the premises they were built under. I understand that in this paper authors evaluate a certain property but how do you know that ResNet-18 does not have an effect on this?
-- To understand table 1 better, is Resnet = Resnet-18 and CapsNets= ResNet-18 backbone + Capsules?

Improvements:
-- Generalise this approach to other proposed CapsNets architectures, beyond Dynamic Routing, such as VarCaps (De Sousa et al. AAAI 2020, StarCaps (Ahmed et al.) etc
-- Use the architecture without a ResNet backbone, but just a single CNN layer in the input
-- SmallNorb could be used to see whether the novel viewpoint generalisation is affected and to what extent.
--Along the same lines Affnist could also be used to see the effect on affine transformations (along with or instead of Cifar10 that was presented in the appendix)

Typos:
--Line 3 Section 4 "(see Equation (??))"

---

> ### Author Response · Authors · 2020-11-18
> **Response to Review 1**
>
> We would like to thank the reviewer for the thoughtful comments. Our responses are as follows:
>
> 1.) Why we use ResNet backbone in CapsNet? Does ResNet backbone has an effect on CapsNet?
>
> CapsNet [1] shows many advantages over CNNs. However, It achieves an unsatisfying performance on complex datasets (e.g., CIFAR10). To make it comparable to SOTA CNNs, many works apply the ResNet backbones to extract more accurate primary capsules. In many versions of CapsNets, the routing is built on a strong backbone, e.g., Self-Routing CapsNet [2],  Inverted dot-product attention routing CapsNet [3].
>
> Hence, we take the backbone-base CapsNets into considerations. In addition, we also did the experiments on the CapsNet on the standard convolutional layers (i.e., the original CapsNet architecture [1]). Please see our response to the fourth question for more details.
>
> 2.) Yes. In Table 1, ResNet = ResNet-18, and CapsNets = ResNet-18 backbone + Capsules with Dynamic Routing. The only difference between CapsNets in Table 1 and the original CapsNet [1] is that the backbone is used in our experiments.
>
> 3.) Generalizing our vote attack to CapsNets with different routings:
>
> Please see our general response and our discussion in the last section of the revised submission.
>
> 4.) Applying our Vote-Attack on CapsNet without a ResNet backbone, but just standard convolutional layer (i.e., the original architecture [1]):
>
> In our paper, we did the experiment. In Section 5.3, we apply our Vote-attack to bypass the class-conditional capsule reconstruction based detection method in CapsNet. In this experiment, the original CapsNet architecture [1] is used (namely, the CapsNet with standard convolutional layers, no backbone). We consider two types of attacks: 1) Detection-aware Attack: We formulate our vote attack by taking the detection into consideration as in Equation (5) and (6). 2) ***Detection-agnostic Attack***: We apply our Vote-Attack to attack CapsNets directly. In this case, it corresponds to the exact experiments suggested by the reviewer.
>
> The performance is shown in Table 4, where two scores are reported in each cell, namely, attack success rate and undetected rate. We can observe that Vote-Attack achieves a much higher success rate than Caps-Attack. More results on different datasets can also be found in Table 9 in Appendix E. The result clearly shows that our Vote attack is also more effective than Caps-Attack on the original CapsNet architecture. The discussion is shown in the last paragraph of Section 5.
>
> 5.) More experiments on AffNIST and SmallNorb datasets:
>
> We add additional experiments on AffMNIST in Section 5.1, where the original CapsNet architecture is used [1]. Our Vote-Attack is also more effective than Caps-Attack.
>
> EM-CapsNet [5] shows better generalization ability to novel viewpoints on SmallNorb dataset. We leave similar explorations on more datasets, more challenging tasks, and more CapsNet versions in future work.
>
> [1] Sabour, Sara, Nicholas Frosst, and Geoffrey E. Hinton. "Dynamic routing between capsules." Advances in neural information processing systems. 2017.
>
> [2] Hahn, Taeyoung, Myeongjang Pyeon, and Gunhee Kim. "Self-Routing Capsule Networks." Advances in Neural Information Processing Systems. 2019.
>
> [3] Tsai, Yao-Hung Hubert, et al. "Capsules with Inverted Dot-Product Attention Routing." International Conference on Learning Representations. 2020.

---

> > ### Comment · AnonReviewer1 · 2020-11-20
> > **Further comments to review**
> >
> > Thanks for the responses. I am happy that you have considered the recommendations and that you have amended the paper accordingly. I think the paper is mature enough to be accepted, and am looking forward to seeing your future work on Capsules.

---

> > > ### Author Response · Authors · 2020-11-21
> > > **Thanks for your encouraging words**
> > >
> > > Your encouraging words are greatly appreciated. We will keep exploration on this topic.

---

### Author Response · Authors · 2020-11-18
**General Response:**

We thank the reviewers for their time and valuable feedback. Following the reviewer’s suggestions, we include more experiments and add the corresponding analysis in our paper. The main changes are

1. In Section 5.1, we add an experiment on the AffNIST dataset where CapsNet built on standard convolutional layers is used, as in [1]. More details can be found in Appendix D.

2. In Section 5.3, we visualize the adversarial images created by Caps-Attack and our Vote-Attack. More figures and the experimental details can be found in Appendix F.

3. We add a paragraph in the last section to discuss the generalization of our Vote-Attack to other versions of CapsNets.

A general question shared by reviewers is about the generalization of our attack to other versions of CapsNets:
To our knowledge, our Vote Attack is the first attack method designed specifically for CapsNet. Hence, we focus on one of the most typical CapsNet architecture, namely, the CapsNet with Dynamic Routig [1].

We also realize the generalization of our attack to different CapsNet versions, as we mentioned in the second paragraph in the last section.

The overall idea of directly attacking votes of CapsNet can be useful to attack other versions of CapsNet since similar votes are available there. However, the architectures of different CapsNet versions often differ from each other. E.g., in EM-CapsNet [2] and VarCaps [3], the existence probability of each matrix-capsule is represented by a single activation unit. It is not trivial to apply our attack to them directly.

Hence, we leave further exploration in future work. We will conduct a comprehensive study on the generalization of our attack method in future work.

[1] Sabour, Sara, Nicholas Frosst, and Geoffrey E. Hinton. "Dynamic routing between capsules." Advances in neural information processing systems. 2017.
[2] Hinton, Geoffrey E., Sara Sabour, and Nicholas Frosst. "Matrix capsules with EM routing." International conference on learning representations. 2018.
[3] Ribeiro, Fabio De Sousa, Georgios Leontidis, and Stefanos D. Kollias. "Capsule Routing via Variational Bayes." AAAI. 2020.

---

### Author Response · Authors · 2021-05-27
**Source Code**

Our official PyTorch implementation is available at https://github.com/JindongGu/VoteAttack

---

### Decision · Program_Chairs · 2021-01-07
**Final Decision**

**Decision:**

Accept (Poster)

**Comment:**

This paper studies the robustness of CapsNets under adversarial attacks. It is found that the votes from primary capsules in CapsNets are manipulated by adversarial examples and that the computationally expensive routing mechanism in CapsNets incurs high computational cost. As such, a new adversarial attack is specially designed by attacking the votes of CapsNets without having to involve the routing mechanism, making the method both effective and efficient.

**Strengths:**
  * This is the first work which proposes an attack specifically designed for CapsNets by exploiting their special properties.
  * The proposed vote attack is more effective and efficient than the other attacks originally proposed for CNNs rather than CapsNets.
  * The paper is generally well written.
  * The experimental study is quite comprehensive.
  * The code will be made available to facilitate reproducibility.

**Weaknesses:**
  * The study is mostly for only one type of CapsNets. It is not clear whether the observations in this paper still hold generally for other types of CapsNets even after some additional experiments have been added.
  * The presentation of the paper has room for improvement.

The authors are recommended to proofread the references thoroughly to ensure style consistency such as the consistent use of capitalization, e.g.
  * “Star-caps” -> “STAR-Caps”
  * “ieee symposium on security and privacy (sp)” -> “IEEE Symposium on Security and Privacy (SP)”

Despite its weaknesses especially those pointed out by Reviewer 2, this paper would be of interest to other researchers as it is the first paper that studies adversarial attacks on CapsNets.